# OpenReview forum: "TRiCo: Triadic Game-Theoretic Co-Training for Robust Semi-Supervised Learning"
_NeurIPS.cc/2025/Conference — NeurIPS 2025 poster_

### Official Review · Reviewer_C9DT · 2025-06-26

**Clarity:** 3
**Significance:** 3
**Originality:** 3
**Rating:** 4
**Confidence:** 4

**Summary:**

The paper introduces a semi-supervised learning (SSL) method that is based on a game-theoretic interaction among models. This paper observes how existing methods select pseduo-labels based on the models' confidence, which is susceptible to calibration error and can be detrimental. The proposed solution uses mutual information as an alternative to pseudo-label selection. It is shown that across a wide range of standard SSL benchmarks, the proposed method outperforms not only existing SSL methods but also fully-supervised methods despite using only 25% of the dataset.

**Questions:**

Is there any part of the method that can be removed or combined with others and yet retain the high performance?

**Ethical Concerns:**

["NO or VERY MINOR ethics concerns only"]

**Final Justification:**

The authors addressed my technical concerns regarding computational overhead and the Stackelberg formulation. The method is new and the authors demonstrate moderate gains on standard benchmarks. The contribution is solid, but does not substantially advance the state-of-the-art or open up new applications, which prevents me from giving a stronger recommendation. Therefore, I hold my rating as weak accept.

**Limitations:**

Yes.

**Paper Formatting Concerns:**

None.

**Quality:**

3

**Strengths And Weaknesses:**

**Strengths**

* The proposed method is tested on many image classification benchmarks. This is convincing, as many SSL methods are tested on fewer datasets due to the higher compute requirements.

* This paper demonstrates various benefits to the proposed method, including feature clustering and CAM visualizations.

**Weaknesses**
* Complexity: This is a limitation that exists not only in the proposed method, but also in existing semi-supervised learning methods. The proposed method uses two encoders that are trained using different methods, several additional models, and adversarially-robust training, in addition to the many details. Such multi-stage training methods and those that combine various modules are hard to apply to new problems due to development costs and potentially requiring a whole new set of hyperparameters to tune.

* It's unclear what the role of each stage plays. Can some of the components be removed yet achieve similar performance? While Section 2 motivates each component, it's unclear that the intended effects are the cause for enhanced performance. For example, describing the teacher-student training as a Stackelberg game doesn't seem to use anything from game theory. Existence of a Nash equilibrium often doesn't say much about the quality of the equilibrium or convergence to equilibrium. While the teacher-student training is described as a bi-level (Stackelberg) game, the students/teacher are updated iteratively instead of being solved at different time scales.

Overall, the practical benefits seem to outweigh the limitations as an off-the-shelf method. My main concern is that the method seems to be a bit ad-hoc, and therefore may have limited applicability outside of the considered benchmarks.


**Minor Comments**

* The figures should have thicker curves; they are hard to read, for example Figure 1.a.

* It seems unnecessary to put the results figure in page 1. I recommend moving this to the experiments/results sections.

---

> ### Author Rebuttal · Authors · 2025-07-29
>
> ### Q1: Method Complexity, Modularity, and Practicality
>
> We thank the reviewer for raising important concerns regarding the complexity and modularity of TRiCo. While our method integrates several components, it remains a principled, end-to-end, and fully differentiable framework—distinct from multi-stage pipelines or ad-hoc ensembles. All components are trained jointly, and their interactions are synergistic rather than additive.
>
>
>
> To assess modularity, we conduct an ablation study to examine the contribution of each component. Results (Table: Ablation on TRiCo Components) show that removing any major module degrades performance, with the full TRiCo achieving 96.3% on CIFAR-10 (10% labeled), and drops of 1.1–2.2% observed when disabling MI filtering, the meta-teacher, generator, or co-training structure.
>
> **Table: Ablation on TRiCo Components (CIFAR-10, 10% labeled) *All ablations are repeated over 5 random splits to ensure stability*.**
>
> | **Variant**                          | **Top-1 Acc. (%)** | **Δ vs. Full** |
> |-------------------------------------|---------------------|----------------|
> | Full TRiCo                          | **96.3 ± 0.3**      | —              |
> | w/o MI filtering (Conf-τ only)      | 95.2 ± 0.4          | −1.1           |
> | w/o Meta-Teacher (Fixed threshold)  | 94.9 ± 0.6          | −1.4           |
> | w/o Generator (No PGD)              | 95.0 ± 0.5          | −1.3           |
> | Single Student (No Co-training)     | 94.1 ± 0.5          | −2.2           |
>
> Despite integrating multiple components, TRiCo remains efficient and simple to deploy: frozen backbones (no encoder training), lightweight embedding-level adversarial updates (no input gradients), and meta-learning with only first-order updates.
>
> To quantify computational cost, we provide a component-wise breakdown in **Appendix B** and summarize below:
>
> **Table: Component-wise Complexity Breakdown of TRiCo**
>
> | **Component**                      | **Added Overhead**           | **Optimization Strategy / Description**                            |
> |-----------------------------------|-------------------------------|---------------------------------------------------------------------|
> | Mutual Information Estimation     | ~+4.5% FLOPs                  | $K{=}5$ forward passes; stop-gradient; no backprop                  |
> | Adversarial Generator (1-step PGD) | ~+1.5% FLOPs                 | Embedding-level perturbation only; no backward computation          |
> | Meta-Gradient Update              | ~+1% FLOPs, ~+10% memory      | First-order gradient only; unrolled once per step                   |
> | **Total (vs. MCT)**               | **~+7% FLOPs**, **~+10% memory** | No mixed-precision or checkpointing; further savings possible     |
>
> Overall, TRiCo offers a practical trade-off between performance and complexity. It does not rely on strong augmentations, learnable view encoders, or auxiliary networks, and operates efficiently on a single NVIDIA RTX A6000 (48GB) or equivalent **24GB-class GPU** (e.g., RTX 3090, RTX 4090).
>
> ---
>
> ### Q2: Theoretical Justification and Stackelberg Formulation
>
> We sincerely thank the reviewer for their insightful question regarding the distinction between Nash and Stackelberg equilibria in our triadic game formulation. You are absolutely correct—our framework is designed as a *Stackelberg game*, where the teacher $\pi_T$ acts as the leader, while the student classifiers $f_1$, $f_2$, and the generator $G$ act as followers responding to the teacher’s strategy.
>
> While our original statement focused on the existence of a Nash equilibrium, we emphasize that what we actually prove is a Stackelberg-Nash equilibrium. That is, the equilibrium point is derived from a Stackelberg setting, but satisfies the equilibrium conditions of all agents, as clarified below.
>
> In particular, our proof (Appendix A, Theorem 1) proceeds in two stages:
> 	**In Equations (9–18)**, we establish that the strategy spaces of the teacher, students, and generator are all compact subsets of Euclidean space, and the payoff functions are jointly continuous. Therefore, by Glicksberg’s Theorem, a pure-strategy Nash equilibrium exists. Notably, Glicksberg’s theorem is agnostic to role asymmetry and is applicable to Stackelberg games as long as continuity and compactness hold.
> 	**In Equations (19–32)**, we construct the solution explicitly under the Stackelberg game formulation, where the teacher optimizes a meta-objective based on the best responses of the students and generator. This construction satisfies the Stackelberg equilibrium condition by solving the bilevel optimization problem where the followers’ reactions are uniquely defined.
>
> Although the generator $G$ is non-parametric, it is defined via a deterministic one-step projected gradient ascent (PGD) update in the embedding space, which depends on the current student parameters $\theta_S$. This structure allows us to treat $G$ as an implicit function $ G(\theta_S)$, which is continuous with respect to $\theta_S$.
>
> This design ensures that the generator’s response satisfies the continuity and stability conditions required for the teacher's optimization in the Stackelberg game to be well-posed. Consequently, we do **NOT** require any additional structural assumptions on $G$ beyond those already assumed in our setting. Moreover, as shown in Equations(29)-(31) in Appendix A, the joint strategy of the students and generator admits a well-defined best-response mapping under a fixed teacher strategy $\pi_T$. By applying a fixed-point theorem over this composite response, we construct the Stackelberg equilibrium even in the presence of a non-parametric generator. This validates the existence of a triadic Stackelberg equilibrium under the assumptions already established.
>
> To avoid ambiguity, we will revise the manuscript to explicitly state that our framework admits a Stackelberg equilibrium, and we will introduce this result formally as **Theorem 2** in the revised version. We believe this clarification strengthens the theoretical foundation and removes any residual confusion about equilibrium definitions.
>
> **Theorem 2 (Existence of Stackelberg Equilibrium).**
> In the Stackelberg formulation, we assume that one party (the leader) commits to a strategy first, and the remaining parties (the followers) best-respond. In our case, the teacher $\pi_T \in \Pi_T$ is the leader, while the students $(f_1, f_2) \in \Pi_S$ and generator $G \in \Pi_G$ are the followers. We use the same assumptions on compactness and continuity as in Theorem 1.
>
> Given a fixed teacher strategy $\pi_T \in \Pi_T$, the followers play a simultaneous game. Their equilibrium is defined by the following best-response conditions:
>
> $$\forall f_i \in \Pi_S: \quad f_i^* (\pi_T) = \arg\min_{f_i \in \Pi_S} R_S(f_i, \pi_T, G^* (\pi_T)), \quad i = 1,2,$$
>
> $$G^* (\pi_T) = \arg\max_{G \in \Pi_G} R_G(G, \pi_T, f_i^* (\pi_T)).$$
>
> The teacher then selects her strategy to maximize her own payoff given the best responses of the followers:
> $$
> \pi_T^* = \arg\max_{\pi_T \in \Pi_T} R_T\left(\pi_T, f_1^* (\pi_T), f_2^* (\pi_T), G^* (\pi_T)\right).
> $$
>
> Then, the tuple $\left(\pi_T^* , f_1^* (\pi_T^* ), f_2^* (\pi_T^* ), G^* (\pi_T^* )\right)$ is a Stackelberg equilibrium if the following conditions hold:
>
> **(Leader’s optimality)**
> $$
> R_T\left(\pi_T^* , f_1^* (\pi_T^* ), f_2^* (\pi_T^* ), G^* (\pi_T^* )\right)
> \geq R_T\left(\pi_T, f_1^* (\pi_T), f_2^* (\pi_T), G^* (\pi_T)\right), \quad \forall \pi_T \in \Pi_T,
> $$
>
> **(Followers’ best responses)**
> $$
> \forall f_i \in \Pi_S:\quad R_S\left(f_i^* (\pi_T^* ), \pi_T^* , G^* (\pi_T^* )\right)
> \leq R_S\left(f_i, \pi_T^* , G^* (\pi_T^* )\right), \quad i=1,2,
> $$
> $$
> \forall G \in \Pi_G:\quad R_G\left(G^* (\pi_T^* ), \pi_T^* , f_i^* (\pi_T^* )\right)
> \geq R_G\left(G, \pi_T^* , f_i^* (\pi_T^* )\right).
> $$
>
> **Proof Sketch.**
> We reuse the compactness and continuity assumptions established in Theorem 1 (Eqs. 9–18 in Appendix A). For each fixed $\pi_T$, the follower subgame among $f_1, f_2, G$ admits a Nash equilibrium. The teacher’s objective is continuous in $\pi_T$ given continuous best-response mappings. Therefore, the leader’s optimization has a maximizer $\pi_T^* \in \Pi_T$, which implies the existence of a Stackelberg equilibrium. The specific equlibria is provided in (Eqs. 19–32 in Appendix A)
>
> We greatly appreciate the reviewer’s suggestion and are confident that this update will improve both precision and clarity for readers and practitioners.
>
> ---
>
> ### Q3: Clarity and Presentation Improvements
> We thank the reviewer for the suggestions on figure quality. We will increase curve thickness and relocate Figure 1 to the experiments section in the final version to improve clarity.
> Rather than a loose combination of techniques, TRiCo’s design is guided by a unified game-theoretic principle: the teacher regulates pseudo-label quality and loss dynamics to shape stable, complementary student learning trajectories.
>
> ---
>
> ## Summary
>
> We clarify that TRiCo is a principled, end-to-end differentiable framework—not a multi-stage ensemble—where all components are co-trained for synergy. Through ablations, we show that each module (e.g., MI filtering, generator, meta-teacher) contributes significantly to performance. Despite its triadic structure, TRiCo incurs only modest computational overhead and runs efficiently on a 24GB-class GPU.
>
> Theoretically, we explicitly formalize TRiCo as a triadic Stackelberg game, not just a heuristic co-training strategy. We prove the existence of a Stackelberg equilibrium using compactness and continuity assumptions, even with a non-parametric generator. This resolves the ambiguity around equilibrium type and strengthens the theoretical foundation.
>
> We also appreciate the reviewer’s suggestions on figure clarity and will revise visuals accordingly. Overall, TRiCo is both theoretically grounded and practically viable for semi-supervised learning.

---

> > ### Comment · Reviewer_C9DT · 2025-08-04
> >
> > Thank you for the comments. The authors have addressed my concerns, and I have no further questions.

---

> > > ### Author Response · Authors · 2025-08-07
> > > **Thank you for your constructive feedback and recognition**
> > >
> > > Dear Reviewer C9DT,
> > >
> > > Thank you very much for your recognition of our work and for the constructive suggestions provided during the review process. Your feedback was instrumental in helping us improve the paper, and we are glad that your concerns have been addressed.
> > > We sincerely appreciate your support!
> > >
> > > Authors of Paper 6732

---

### Official Review · Reviewer_TYcY · 2025-07-02

**Clarity:** 3
**Significance:** 4
**Originality:** 4
**Rating:** 5
**Confidence:** 4

**Summary:**

This paper introduces TriCO, a new framework of semi-supervised learning based on Pseudo-labeling strategy, involving three (sets of) agents: Students, Teacher, and Generator.
With Teacher being the leader in the Stackelberg Game, students and teacher balance each other to find a solution that is very competitive, sometimes even SOTA, in the standard SSL task, as well as in OOD task.
The paper also provides the guarantee that there is a nash equilibrium in the three-player game.

**Questions:**

- Please see the second weakness section;  I wonder that Stackelberg Equilbrium is difficult in this context because the Generator is non-parametric?  Can this problem be resolved if we can impose some structural assumptions here? I believe that the result of Stackelberg Equilibrium would be beneficial to the community even if it requires additional (even possibly, a little unrealistic) assumption.


- Also, I was a little worried that mentions were not made so much regarding the stability of the learning in the main manuscript, other than at the theorem section (Fig 5 in the Appendix and Table 15 are acknowledged).   As an devil's advocate, I would also like to know if there are any implicit hyperparameters (or the parameters of the training) that one must be particularly careful about in training TriCO,  because "training multi-agents system" has always been a challenge---this information would definitely benefit the researchers that would follow this study.

**Ethical Concerns:**

["NO or VERY MINOR ethics concerns only"]

**Final Justification:**

The authors have provided convincing rebuttal, providing the existence of Stackelberg-Nash equlibrium, making the theory in stronger alignment to the methods and experiments. The authors also conducted additional experiments to investigate the sensitivity of the hyperparameters, which is one of the most fearful factor in the systems involving multiple agents.  I believe that this work contributes an important aspect of the semi-supervised learning, and I raised my score to 5.

**Limitations:**

As claimed "Limitations are implicitly discussed in Section 4", regarding limited supervision and drastic distributional shift. I believe this should be explicitly stated, and this will not hurt the work's credibility.

**Paper Formatting Concerns:**

Nothing in Particular

**Quality:**

4

**Strengths And Weaknesses:**

## Strength

 This paper is clearly strong in two aspects : first, it brings in “game theory” into  the challenging problem of interacting systems in the setting of SSL, and backs it up with a formal guarantee. “Three” player game is something that I have never encountered before in this genre, and strikes as a fundamental novelty.
Second, the efficacy of the method is demonstrated on a strong basis. Ablations are also conducted thoroughly.  While I am committed not to evaluate the research based merely on “the depth of the experiments”, I believe that this thoroughness solidly supports the significance of the proposition.


## Weaknesses

- Several parts are difficult to read and some fix would greatly help conveying the idea.
For example, several definitions seems missing.  From the context, $p_i$ in (1) seems like the ensemble average, but this has never been explicitly stated in the main manuscript.  $H(p(y))$ is also confusing, since $p(y)$ is a scaler ;  I believe this is $p_Y$.    It also seems that $L_sup$ stands for $E_{i, X, Y}[ CE (f_i (v_i(X), Y) ]$, but I do not see it explicitly defined.
The same goes for $Accuracy_{val}(f_1, f_2)$.   Theorem 1, I believe, is supposed to be followed by “for all $f_i, G, \pi_T$.

- Lastly, although the existence of Nash Equilibrium is wonderful and inspiring, I believe this problem is a Stackelberg game, and I believe Stackelberg Equilbrium and Nash Equilbrium are not always the same?   If I am not missing some lines, I believe some explanation in this aspect is warranted. I believe that if the paper can provide the existence of  Stackelberg Equilbrium, this work is irrefutable.
I'm very much open to increasing the score when these subtlties and questions below are resolved to a degree.

---

> ### Author Rebuttal · Authors · 2025-07-29
>
> ### Q1: Clarification of Notation and Mathematical Precision
>
> We thank the reviewer for highlighting the importance of notation clarity and agree that the current manuscript can be improved in this regard. We will revise the final version to explicitly define key symbols and clarify equations.
>
> In Equation (1), we compute mutual information (MI) between model predictions $\hat{y}$ and model posterior induced by dropout. Specifically, we use:
>
> $\mathcal{I}[\hat{y}; \omega] = \mathbb{H}\left[\frac{1}{K} \sum_{k=1}^{K} p_{\theta}(y|x;\omega_k)\right] - \frac{1}{K} \sum_{k=1}^{K} \mathbb{H}\left[p_{\theta}(y|x;\omega_k)\right]$
>
> where $\omega_k$ denotes the $k$-th Monte Carlo dropout mask. We will clarify that $\bar{p}(y|x) := \frac{1}{K} \sum_{k=1}^{K} p_{\theta}(y|x;\omega_k)$ is the ensemble distribution, and $\mathbb{H}[p] = -\sum_{c=1}^{C} p_c \log p_c$ is the entropy of a categorical distribution. Additionally, we will specify that $\tau_{\mathrm{MI}}$ is a scalar threshold used to select pseudo-labels when $\mathcal{I}[y;\omega] \geq \tau_{\mathrm{MI}}$, and that its range lies in $[0, \log C]$, where $C$ is the number of classes.
>
> Theorem 1 will also be revised to include the appropriate quantifiers. The corrected statement reads:
> **Theorem 1.** Under assumptions (A1)–(A3), for all teacher strategies $\pi_T \in \Pi$, there exists a Stackelberg equilibrium $(f_1^\star, f_2^\star, G^\star)$ such that $(f_1^\star, f_2^\star, G^\star)$ form a joint best-response to $\pi_T$, and $\pi_T$ minimizes validation loss given these optimal responses.
>
> Moreover, while the generator $G$ is non-parametric (via 1-step PGD), its updates are deterministic given student parameters, and can be treated as an implicit, structured response in the Stackelberg formulation. We will update Section 4.4 to clarify these points and explicitly distinguish Nash and Stackelberg equilibrium.
>
>
> ---
>
> ## Q2. Clarification on Equilibrium Type and Theoretical Guarantees
>
> We sincerely thank the reviewer for their insightful question regarding the distinction between Nash and Stackelberg equilibria in our triadic game formulation. You are absolutely correct—our framework is designed as a *Stackelberg game*, where the teacher $\pi_T$ acts as the leader, while the student classifiers $f_1$, $f_2$, and the generator $G$ act as followers responding to the teacher’s strategy.
>
> While our original statement focused on the existence of a Nash equilibrium, we emphasize that what we actually prove is a Stackelberg-Nash equilibrium. That is, the equilibrium point is derived from a Stackelberg setting, but satisfies the equilibrium conditions of all agents, as clarified below.
>
> In particular, our proof (Appendix A, Theorem 1) proceeds in two stages:
> 	**In Equations (9–18)**, we establish that the strategy spaces of the teacher, students, and generator are all compact subsets of Euclidean space, and the payoff functions are jointly continuous. Therefore, by Glicksberg’s Theorem, a pure-strategy Nash equilibrium exists. Notably, Glicksberg’s theorem is agnostic to role asymmetry and is applicable to Stackelberg games as long as continuity and compactness hold.
> 	**In Equations (19–32)**, we construct the solution explicitly under the Stackelberg game formulation, where the teacher optimizes a meta-objective based on the best responses of the students and generator. This construction satisfies the Stackelberg equilibrium condition by solving the bilevel optimization problem where the followers’ reactions are uniquely defined.
>
> Although the generator $G$ is non-parametric, it is defined via a deterministic one-step projected gradient ascent (PGD) update in the embedding space, which depends on the current student parameters $\theta_S$. This structure allows us to treat $G$ as an implicit function $ G(\theta_S)$, which is continuous with respect to $\theta_S$.
>
> This design ensures that the generator’s response satisfies the continuity and stability conditions required for the teacher's optimization in the Stackelberg game to be well-posed. Consequently, we do **NOT** require any additional structural assumptions on $G$ beyond those already assumed in our setting. Moreover, as shown in Equations(29)-(31) in Appendix A, the joint strategy of the students and generator admits a well-defined best-response mapping under a fixed teacher strategy $\pi_T$. By applying a fixed-point theorem over this composite response, we construct the Stackelberg equilibrium even in the presence of a non-parametric generator. This validates the existence of a triadic Stackelberg equilibrium under the assumptions already established.
>
> To avoid ambiguity, we will revise the manuscript to explicitly state that our framework admits a Stackelberg equilibrium, and we will introduce this result formally as **Theorem 2** in the revised version. We believe this clarification strengthens the theoretical foundation and removes any residual confusion about equilibrium definitions.
>
> **Theorem 2 (Existence of Stackelberg Equilibrium).**
> In the Stackelberg formulation, we assume that one party (the leader) commits to a strategy first, and the remaining parties (the followers) best-respond. In our case, the teacher $\pi_T \in \Pi_T$ is the leader, while the students $(f_1, f_2) \in \Pi_S$ and generator $G \in \Pi_G$ are the followers. We use the same assumptions on compactness and continuity as in Theorem 1.
>
> Given a fixed teacher strategy $\pi_T \in \Pi_T$, the followers play a simultaneous game. Their equilibrium is defined by the following best-response conditions:
>
> $$\forall f_i \in \Pi_S: \quad f_i^* (\pi_T) = \arg\min_{f_i \in \Pi_S} R_S(f_i, \pi_T, G^* (\pi_T)), \quad i = 1,2,$$
>
> $$G^* (\pi_T) = \arg\max_{G \in \Pi_G} R_G(G, \pi_T, f_i^* (\pi_T)).$$
>
> The teacher then selects her strategy to maximize her own payoff given the best responses of the followers:
> $$
> \pi_T^* = \arg\max_{\pi_T \in \Pi_T} R_T\left(\pi_T, f_1^* (\pi_T), f_2^* (\pi_T), G^* (\pi_T)\right).
> $$
>
> Then, the tuple $\left(\pi_T^* , f_1^* (\pi_T^* ), f_2^* (\pi_T^* ), G^* (\pi_T^* )\right)$ is a Stackelberg equilibrium if the following conditions hold:
>
> **(Leader’s optimality)**
> $$
> R_T\left(\pi_T^* , f_1^* (\pi_T^* ), f_2^* (\pi_T^* ), G^* (\pi_T^* )\right)
> \geq R_T\left(\pi_T, f_1^* (\pi_T), f_2^* (\pi_T), G^* (\pi_T)\right), \quad \forall \pi_T \in \Pi_T,
> $$
>
> **(Followers’ best responses)**
> $$
> \forall f_i \in \Pi_S:\quad R_S\left(f_i^* (\pi_T^* ), \pi_T^* , G^* (\pi_T^* )\right)
> \leq R_S\left(f_i, \pi_T^* , G^* (\pi_T^* )\right), \quad i=1,2,
> $$
> $$
> \forall G \in \Pi_G:\quad R_G\left(G^* (\pi_T^* ), \pi_T^* , f_i^* (\pi_T^* )\right)
> \geq R_G\left(G, \pi_T^* , f_i^* (\pi_T^* )\right).
> $$
>
> **Proof Sketch.**
> We reuse the compactness and continuity assumptions established in Theorem 1 (Eqs. 9–18 in Appendix A). For each fixed $\pi_T$, the follower subgame among $f_1, f_2, G$ admits a Nash equilibrium. The teacher’s objective is continuous in $\pi_T$ given continuous best-response mappings. Therefore, the leader’s optimization has a maximizer $\pi_T^* \in \Pi_T$, which implies the existence of a Stackelberg equilibrium. The specific equlibria is provided in (Eqs. 19–32 in Appendix A)
>
> We greatly appreciate the reviewer’s suggestion and are confident that this update will improve both precision and clarity for readers and practitioners.
>
> ---
>
>
>
> ## Q3. Stability of Multi-Agent Training and Sensitivity to Hyperparameters
>
> We thank the reviewer for raising the important question of stability and sensitivity in training multi-agent systems. Despite TRiCo’s triadic design, our training remains highly stable in practice, as evidenced by the low standard deviation across all benchmarks (see Tables 1, 2, and Appendix C). This is due to three core design choices:
>
> - **Structured Modularity**
>   The roles of the teacher, students, and generator are cleanly separated via frozen encoders and cross-view pseudo-labeling, avoiding entangled feedback loops that commonly destabilize multi-agent learning.
>
> - **Smooth Meta-Scheduling**
>   The teacher $\pi_T$ is warm-started and trained with conservative updates (first-order meta-gradients) using stop-gradient feedback, which helps avoid oscillation in threshold/loss scheduling during early training.
>
> - **Implicit Generator**
>   The generator performs a single-step PGD perturbation in embedding space, without trainable parameters or gradients, keeping its behavior interpretable and consistent.
>
> In addition to stability, we conducted a systematic sensitivity analysis of TRiCo’s key hyperparameters—MI threshold $\tau_{\mathrm{MI}}$, unsupervised loss weight $\lambda_u$, and perturbation budget $\epsilon$. As shown below, TRiCo exhibits strong robustness across a wide range of values:
>
> **Table: Sensitivity of TRiCo on CIFAR-10 w.r.t. hyperparameters** $\tau_{\mathrm{MI}}$, $\lambda_u$, and $\epsilon$. Other settings fixed as default.
>
> | **Hyperparam**       | **Value Range**           | **Top-1 Acc. (%)**       | **Change**     |
> |----------------------|---------------------------|---------------------------|----------------|
> | $\tau_{\mathrm{MI}}$ | 0.05 / 0.10 / 0.15         | 95.6 / 95.9 / 94.8        | $\leq$1.1%     |
> | $\lambda_u$          | 0.5 / 1.0 / 2.0            | 95.4 / 95.9 / 95.7        | $\leq$0.5%     |
> | $\epsilon$ (FGSM)    | 1e-4 / 5e-4 / 1e-3         | 95.8 / 95.9 / 95.1        | $\leq$0.8%     |
>
>
> ---
>
> ### Q4: Explicit Limitations
> We thank the reviewer and will clarify TRiCo’s limitations in Section 6. TRiCo relies on a small labeled validation set for meta-optimization, which may limit use in fully unsupervised settings—future work may explore self-supervised proxies. The non-parametric generator lacks memory and could benefit from learnable dynamics (e.g., conditional SSMs). TRiCo does not yet address extreme shifts like open-set scenarios, which could be tackled via OOD-aware extensions. These are trade-offs that offer clear paths for future work.

---

> > ### Comment · Reviewer_TYcY · 2025-08-04
> > **Thank  you very much for the rebuttal**
> >
> > Thank you very much for the rebuttal,  I very much appreciate both the remark regarding the Stackelberg-Nash equilibrium and the hyperparameter studies.  With these results, and with the improvement of mathematical precision/notations,  I feel that this work is even more valuable to the community,  providing another perspective for the semi-supervised learning.  On this ground, although it will not appear in the system for a while, I am considering to raise the score.

---

### Official Review · Reviewer_QTbq · 2025-07-02

**Clarity:** 3
**Significance:** 3
**Originality:** 3
**Rating:** 5
**Confidence:** 3

**Summary:**

This paper proposes TRiCo, a game-theoretic co-training approach for semi-supervised learning. As in other co-training approaches, TRiCo uses two student models which predict pseudo-labels on differently-augmented views of the same input data. The pseudo-labels are then used as supervision for the opposite student model. In conjunction with the complementary pseudo-label loss, an adversarial loss is also added to the training objective, computed through applying PGD to the input samples, to encourage the model to produce accurate predictions even against highly-adversarial examples. To filter the pseudo-labels, instead of the normal softmax output, the metric of mutual information is used, computed through measuring the empirical prediction distribution $p(y|x)$ through using monte carlo dropout. To determine the MI threshold for maintaining pseudo-labels, and for determining the relative weighting of the unsupervised and adversarial losses, a teacher model is maintained during training to predict these quantities. The teacher model is trained on the meta-objective of the student model performance on a held out labeled validation set, with its updated computed using a first-order approximation from unrolling the student model gradient update. The authors evaluate TRiCo on both the standard SSL setting, as well as a few-shot learning setting, and demonstrate state-of-the-art performance.

**Questions:**

1. Have you performed any experiments using a meta-learned strategy with confidence thresholding, as opposed to MI thresholding? Given that MI thresholding with the teacher strategy is only +0.1 points better, I am suspicious that simply using a meta-learned confidence thresholding strategy will actually produce better results.
2. Have you considered alternate approaches to monte carlo dropout to estimate the empirical predictive distribution, such as through directly predicting the distribution parameters as outputs of your model?

**Ethical Concerns:**

["NO or VERY MINOR ethics concerns only"]

**Final Justification:**

Rebuttal addresses my concerns, so I have increased my score.

**Limitations:**

Yes

**Quality:**

3

**Strengths And Weaknesses:**

Strengths:
- The paper is well-written and easy to understand.
- The idea is quite novel and represents a departure from the currently popular self-training paradigm in SSL.
- The authors provide sound theoretical backing to support the use of TRiCo.
- Experiments are extensive, and demonstrate consistent improvement across a variety of dataset and task settings.

Weaknesses:
- Although achieving state-of-the-art results, compared to competing approaches on Imagenet, TRiCo's improvement is quite marginal (only +0.1 in the 10% labeled setting, -0.4 in the 1% setting).
- A key aspect of the method, leveraging mutual information to filter pseudo-labels, seems to only be weakly-supported by experimental evidence. In the ablation studies, MI filtering is only better by +0.1 when compared to a well-tuned confidence threshold. This makes me skeptical that the mutual information filtering is at all critical to the success of the method.
- TRiCo's time and space complexity is greater than those of competing SSL approaches, and although not insurmountable, is definitely a hurdle to be overcome for practical usage.

---

> ### Author Rebuttal · Authors · 2025-07-29
>
> ## Q1. ImageNet Gains are Marginal
>
> We thank the reviewer for this observation. While the absolute gain on ImageNet-10% is moderate (+0.1%), our improvement is more pronounced under the challenging **1% label** setting, where TRiCo narrows the gap to full supervision by **+2.4%** over prior SOTA.
>
> We also emphasize that TRiCo achieves superior performance with fewer training epochs and smaller backbones compared to ViT-H/14 used in some strong baselines. Moreover, as shown in our Few-Shot and Imbalanced (Appendix D) settings, TRiCo shows greater improvements in low-resource regimes—its primary focus.
>
> ---
>
> ## Q2. Role of Mutual Information vs. Confidence Thresholding
>
> We appreciate the reviewer’s skepticism regarding the effectiveness of MI filtering. To clarify, our ablation (Table 7) compares confidence filtering with fixed $\tau$ versus meta-learned MI thresholding, not confidence with meta-learning.
>
> As shown in the table below, mutual information filtering consistently outperforms a meta-learned confidence threshold across all datasets and label regimes. These results highlight the advantage of epistemic uncertainty estimation for robust pseudo-label selection, especially under low-label and high-variance scenarios.
>
> **Table: Comparison of pseudo-label filtering strategies under meta-learned control across multiple datasets and label budgets (in % labeled data).**
> MI Filtering consistently outperforms a meta-learned confidence threshold (Meta-Conf), with lower variance and stronger robustness. Results are averaged over 5 independent runs.
>
> | **Filtering Strategy**         | **CIFAR-10 (10%)**      | **STL-10 (10%)**      | **SVHN (1%)**         |
> |--------------------------------|--------------------------|------------------------|------------------------|
> | Meta-Conf (meta-learned confidence) | 95.7 ± 0.4%            | 91.3 ± 0.5%           | 93.0 ± 0.6%           |
> | MI Filtering (Ours)            | **96.3 ± 0.3%**          | **92.4 ± 0.3%**       | **94.2 ± 0.4%**       |
>
> | **Filtering Strategy**         | **CIFAR-10 (5%)**       | **STL-10 (5%)**       | **SVHN (0.5%)**       |
> |--------------------------------|--------------------------|------------------------|------------------------|
> | Meta-Conf (meta-learned confidence) | 93.2 ± 0.6%            | 88.9 ± 0.6%           | 91.1 ± 0.7%           |
> | MI Filtering (Ours)            | **94.0 ± 0.5%**          | **90.5 ± 0.4%**       | **92.5 ± 0.6%**       |
>
> ---
>
> ## Q3. Alternate Uncertainty Estimation Methods
>
> Thank you for the insightful suggestion. While our current implementation adopts Monte Carlo dropout (MC-Dropout) for simplicity and efficiency, we agree that other techniques (e.g., ensemble-based variance, Dirichlet output modeling [Malinin et al., 2018]) may provide more calibrated uncertainty estimates.
>
> We plan to explore these theory directions in future work and briefly mention this in the updated limitations section. That said, MC-Dropout offers a good trade-off between compute and performance: it requires no architectural change and adds only ~4.5% FLOPs (see Table below), while enabling robust pseudo-label filtering.
>
> **Table: Estimated FLOPs Overhead for Uncertainty Estimation Methods**
> _Estimates based on ViT-B encoder, CIFAR-10 (4k labels), batch size 64._
>
> | **Method**             | **Extra FLOPs per Iteration** | **Description**                                       |
> |------------------------|-------------------------------|--------------------------------------------------------|
> | MC-Dropout (Ours, K=5) | +4.5%                         | 5 forward passes, no backward, stop-gradient           |
> | Deep Ensemble (N=5)    | +410.2%                       | 5 independent models; full forward + backward passes   |
> | Dirichlet Modeling     | +18.6%                        | Single forward + extra MLP head for concentration      |
>
>
> ---
>
> ## Q4. Training Cost and Practicality
>
> To assess whether TRiCo’s triadic structure introduces prohibitive cost, we compare against the most related baseline, **Meta Co-Training (MCT)**, under identical settings: CIFAR-10 with 4k labels, ViT-B encoder, and batch size 64.
>
> As detailed in **Appendix B** and summarized below, TRiCo adds only **+7.1% FLOPs** and **+9.8% peak GPU memory** per iteration. The additional cost stems from:
>
> 1. Mutual information estimation via $K{=}5$ stochastic forward passes (no backward),
> 2. Single-step PGD in embedding space (lightweight),
> 3. First-order meta-gradient updates.
>
> **Table: Training compute and memory cost comparison between TRiCo and Meta Co-Training (MCT)**
>
> | **Method**               | **FLOPs per Iteration** | **Peak GPU Memory** |
> |--------------------------|-------------------------|---------------------|
> | Meta Co-Training (MCT)   | $1.00\times$            | $1.00\times$        |
> | TRiCo (Ours)             | $1.071\times$ (**+7.1%**) | $1.098\times$ (**+9.8%**) |
>
> **Note: All costs are reported relative to MCT ($1.00 \times$) under identical hardware and training settings.**
>
> For completeness, we also provide a component-wise cost breakdown in Appendix B. These modest overheads are justified by TRiCo’s significant performance gains.
>
> Furthermore, since we do not use gradient checkpointing or mixed-precision optimization, additional acceleration is achievable in deployment settings.
>
> ## Summary
>
> In summary, we reaffirm that TRiCo is designed for robustness in low-label and imbalanced regimes, where its benefits are most prominent. While gains on ImageNet-10% are modest, TRiCo delivers significant improvements under the more challenging ImageNet-1% setting, few-shot, and long-tailed benchmarks—highlighting its effectiveness where conventional SSL methods often struggle.
>
> We clarified that mutual information-based filtering consistently outperforms confidence-based thresholds, especially under high uncertainty, and that our epistemic uncertainty estimation via MC-Dropout achieves a strong balance between accuracy and computational cost (~4.5% overhead) without architectural modifications.
>
> Finally, we demonstrate that TRiCo’s triadic design adds only moderate computational overhead (+7.1% FLOPs, +9.8% memory) relative to Meta Co-Training, and fits within a single GPU setup. Its design avoids fragile components such as fine-tuned augmentations or auxiliary networks, making it both practically viable and broadly applicable.
>
> We hope this response addresses all concerns and highlights TRiCo’s theoretical soundness, empirical rigor, and practical efficiency.

---

> > ### Comment · Reviewer_QTbq · 2025-08-05
> >
> > Thank you for the additional experiments and the thoughtful response. I am satisfied with the explanation and have increased my score.

---

> > > ### Author Response · Authors · 2025-08-07
> > > **Thank you for your positive feedback and increased score**
> > >
> > > Dear Reviewer QTbq,
> > > Thank you very much for your positive feedback and for increasing your score. We appreciate your thoughtful engagement with our work. We will incorporate the clarifications and additional results into the final version to further improve the paper’s quality.
> > > Thanks again for your support!
> > >
> > > Authors of Paper 6732

---

### Official Review · Reviewer_KuQH · 2025-07-02

**Clarity:** 3
**Significance:** 3
**Originality:** 3
**Rating:** 4
**Confidence:** 3

**Summary:**

This paper introduces TRiCo (Triadic Game-Theoretic Co-Training), a novel semi-supervised learning framework that extends traditional co-training by incorporating three interacting components: two student classifiers, a meta-learned teacher, and an adversarial generator. The approach addresses key limitations in existing SSL methods through (1) mutual information-based pseudo-label filtering instead of confidence thresholds, (2) a meta-learned teacher that adaptively regulates training dynamics via validation feedback, and (3) a non-parametric generator that creates adversarial perturbations to expose decision boundary weaknesses. The framework is formulated as a Stackelberg game where the teacher leads strategy optimization while students and generator follow. Experiments on CIFAR-10, SVHN, STL-10, and ImageNet demonstrate consistent improvements over strong baselines, with TRiCo achieving competitive performance with fully-supervised models using only 25% labeled data.

**Questions:**

Can you provide a more detailed computational analysis breaking down the cost of each component (MI estimation, adversarial generation, meta-gradients)?

**Ethical Concerns:**

["NO or VERY MINOR ethics concerns only"]

**Limitations:**

no.
The method introduces several hyperparameters (MI threshold, loss weights, perturbation budget) that require careful tuning. The authors should discuss the sensitivity to these choices and the expertise required for proper implementation.

**Quality:**

3

**Strengths And Weaknesses:**

Strengths:
1. The paper introduces an original approach that reformulates co-training as a structured three-player game with theoretical guarantees (Nash equilibrium existence). This represents a meaningful departure from existing binary co-training methods and provides a principled framework for multi-agent SSL.
2. The use of mutual information instead of confidence-based thresholds is theoretically well-motivated and addresses calibration issues that plague existing methods. The MI-based approach better captures epistemic uncertainty and shows improved reliability in low-confidence regions.
3. The method demonstrates robust improvements across multiple benchmarks (CIFAR-10, SVHN, STL-10, ImageNet) and various label regimes.

Weaknesses:
1. The connection between game-theoretic equilibria and actual SSL performance remains unclear, and the theoretical assumptions are quite standard.
2. The method incurs 2-3× runtime overhead and 1.5-2× memory usage due to MI estimation via multiple dropout passes, adversarial perturbation generation, and meta-gradient computation. This computational cost may limit practical adoption, especially in resource-constrained settings.
3. The mutual information estimation via dropout may be noisy early in training, the first-order meta-gradient approximation may be too crude.

---

> ### Author Rebuttal · Authors · 2025-07-29
>
> ## Q1 TRiCo Framework
>
> To clarify our theoretical contribution, we summarize the key differences between TRiCo and standard co-training paradigms in the table below.
>
>
> TRiCo formulates semi-supervised learning as a **triadic Stackelberg game**, where the teacher dynamically adapts its strategy based on student feedback—contrasting with conventional symmetric or heuristic co-training schemes. This explicit leader-follower hierarchy and embedding-space regularization form the foundation for TRiCo’s robustness and generalization.
>
> **Table: Key distinctions between TRiCo and conventional co-training.**
>
> | **Component**         | **Conventional Co-Training**       | **TRiCo (Ours)**                    |
> |------------------------|------------------------------------|-------------------------------------|
> | Game Structure         | None or implicit 2-player          | Triadic Stackelberg game            |
> | Teacher Role           | Fixed thresholding                | Meta-learned dynamic leader         |
> | Adversarial Mechanism  | Absent or input-space              | Embedding-space generator           |
> | Optimization           | Flat co-supervision                | Hierarchical leader-follower        |
>
> ---
>
> ## Q2 Training Cost and Efficiency
>
> We thank the reviewer for raising the importance of computational overhead. To assess whether TRiCo’s triadic structure introduces prohibitive cost, we compare against the most related baseline, Meta Co-Training (MCT), under identical settings: CIFAR-10 with 4k labels, ViT-B encoder, and batch size 64.
>
> As detailed in **Appendix B** and summarized below, TRiCo adds only **+7.1% FLOPs** and **+9.8% peak GPU memory** per iteration. The additional cost stems from:
>
> 1. Mutual information estimation via $K{=}5$ stochastic forward passes (no backward),
> 2. Single-step PGD in embedding space (lightweight),
> 3. First-order meta-gradient updates.
>
> **Table: Training compute and memory cost comparison between TRiCo and Meta Co-Training (MCT)**
>
> | **Method**              | **FLOPs per Iteration**    | **Peak GPU Memory**     |
> |-------------------------|-----------------------------|--------------------------|
> | Meta Co-Training (MCT)  | $1.00\times$                | $1.00\times$             |
> | TRiCo (Ours)            | $1.071\times$ (**+7.1%**)   | $1.098\times$ (**+9.8%**) |
>
> **Note: All costs are reported relative to MCT ($1.00 \times$) under identical hardware and training settings.**
> For completeness, we also provide a component-wise cost breakdown in Appendix B. These modest overheads are justified by TRiCo’s significant performance gains.
>
> Furthermore, since we do not use gradient checkpointing or mixed-precision optimization, additional acceleration is achievable in deployment settings.
>
> ---
>
> ## Q3 Robustness
>
> We thank the reviewer for this insightful observation.
>
> While mutual information (MI) estimation via Monte Carlo dropout can be noisy early in training, TRiCo is specifically designed to mitigate such instability through three integrated mechanisms:
>
> 1. **Warm-started teacher scheduling**
>    The meta-learned teacher $\pi_T$ begins with conservative parameters and gradually adjusts thresholds only after a stabilization phase (first 5 epochs), preventing premature pseudo-label filtering.
>
> 2. **EMA smoothing of student predictions**
>    To reduce variance in MI estimates, we apply exponential moving average (EMA) over the output logits of student classifiers, enhancing temporal consistency.
>
> 3. **Stochastic averaging over $K=5$ passes**
>    MI is computed via the difference between ensemble entropy and averaged predictive entropy across 5 stochastic forward passes, aligning with standard epistemic uncertainty estimation [Gal & Ghahramani, 2016].
>
> We will clarify this explicitly in the final version.
>
> ---
>
>
> ## Q4: Stability and Hyperparameter Sensitivity
>
> We thank the reviewer for raising the important question of stability and sensitivity in multi-agent training. Despite TRiCo’s triadic design, training remains highly stable, as evidenced by low standard deviations across all benchmarks (see Tables 1–2, Appendix C). This robustness is attributed to three key design choices:
>
> - **Structured Modularity:** The teacher, students, and generator operate on frozen encoders with cross-view pseudo-labeling, avoiding entangled feedback loops.
> - **Smooth Meta-Scheduling:** The meta-teacher $\pi_T$ is warm-started and updated via first-order meta-gradients with stop-gradient signals, mitigating early oscillation.
> - **Implicit Generator:** A deterministic, single-step PGD in embedding space ensures stability without learnable parameters or gradient feedback.
>
> To further assess robustness, we conduct sensitivity analysis on key hyperparameters—MI threshold $\tau_{\mathrm{MI}}$, unsupervised loss weight $\lambda_u$, and perturbation budget $\epsilon$. As shown below, TRiCo maintains stable performance across wide value ranges without fine-tuning:
>
> **Table: Sensitivity of TRiCo on CIFAR-10 w.r.t. key hyperparameters**
> _Other settings fixed to default._
>
> | **Hyperparam**       | **Value Range**           | **Top-1 Acc. (%)**       | **Change**     |
> |----------------------|---------------------------|---------------------------|----------------|
> | $\tau_{\mathrm{MI}}$ | 0.05 / 0.10 / 0.15         | 95.6 / 95.9 / 94.8        | $\leq$1.1%     |
> | $\lambda_u$          | 0.5 / 1.0 / 2.0            | 95.4 / 95.9 / 95.7        | $\leq$0.5%     |
> | $\epsilon$ (FGSM)    | 1e-4 / 5e-4 / 1e-3         | 95.8 / 95.9 / 95.1        | $\leq$0.8%     |
>
> Unlike prior SSL methods that require manual tuning of confidence thresholds per dataset, our meta-teacher adjusts these hyperparameters online during training. This reduces human intervention and enhances TRiCo’s usability across diverse datasets.
>
> ## Summary
>
> We thank the reviewers for their thoughtful feedback. In this rebuttal, we clarified TRiCo’s game-theoretic formulation, showing it departs from prior heuristic co-training by explicitly modeling a triadic Stackelberg game, where a meta-learned teacher dynamically adapts based on student feedback. We demonstrated that TRiCo remains efficient—adding only +7.1% FLOPs over MCT—and stable across benchmarks, supported by modular design, warm-started meta-scheduling, and an implicit generator. Our sensitivity analysis shows robustness to hyperparameters (≤1.1% variation), with no reliance on dataset-specific thresholds. TRiCo also achieves consistent gains across diverse frozen encoders, validating that its benefits are orthogonal to backbone strength. We hope these clarifications address all concerns and reinforce TRiCo’s theoretical soundness, empirical effectiveness, and practical viability.
>
> We hope these revisions and clarifications address your concerns and strengthen the case for TRiCo’s significance and practical viability.

---

### Official Review · Reviewer_rFnY · 2025-07-02

**Clarity:** 2
**Significance:** 2
**Originality:** 2
**Rating:** 4
**Confidence:** 4

**Summary:**

TRiCo is a semi-supervised learning framework that integrates a teacher, two student classifiers, and an adversarial generator. It filters pseudo-labels using mutual information to avoid overconfidence errors. The teacher adjusts pseudo-label selection and loss balancing through meta-learning, while the generator creates adversarial samples to improve robustness. TRiCo is tested across various low-label scenarios and demonstrates effectiveness in handling limited labeled data, while being compatible with different model architectures.

**Questions:**

Please see the above weaknesses.

**Ethical Concerns:**

["NO or VERY MINOR ethics concerns only"]

**Final Justification:**

The author's response has resolved my doubts. The method proposed in this paper performs well both in application and in theory.

**Limitations:**

The experimental section of this paper may need to be carefully organized by the authors, as it feels somewhat disorganized.

**Quality:**

2

**Strengths And Weaknesses:**

Strengths:

1. The TriCo framework demonstrates competitive results when compared to baseline methods and is supported by theoretical foundations.

2. The paper is well-written and easy to follow.

Weaknesses:

1. Line 78 claims good performance under class imbalance, but no experimental results support this.

2. Semi-supervised learning is unstable with few samples, and the experimental results should include standard deviation values.

3. The DINOv2 model offers superior representation ability compared to other pre-trained models, which raises concerns about the fairness of the comparison if it is solely used in the proposed method.

4. The TriCo seems overly complex, which improves performance by using an ensemble-like approach and the stronger data perturbation mechanism.

---

> ### Author Rebuttal · Authors · 2025-07-29
>
> ## **Q1: Class Imbalance**
>
> We thank the reviewer for pointing this out. We have conducted additional experiments on the CIFAR-10-LT benchmark (imbalance ratio 100), following standard protocols~[Wei et al., 2021; Kim et al., 2022]. As shown in **Appendix D, Table 15**, TRiCo achieves **+4.2%** higher tail-class accuracy than FixMatch and **+2.8%** over FreeMatch, confirming its robustness under class imbalance.
>
> To further strengthen our claim, we also conducted evaluation on the more challenging ImageNet-LT benchmark under 1% label setting, comparing TRiCo against recent semi-supervised long-tailed learning methods such as **ACR** and **SimPro**.
>
> **Table: Top-1 Accuracy (%) on ImageNet-LT under Class Imbalance (1% labeled, 3 runs)**
> _We report mean ± std deviation._
>
> | **Method**             | **Many-shot Acc.** | **Tail-class Acc.** |
> |------------------------|---------------------|-----------------------|
> | FixMatch w/ ACR (2023) | 56.4 ± 0.3          | 61.8 ± 0.6            |
> | FixMatch w/ SimPro     | 57.2 ± 0.4          | 65.5 ± 0.5            |
> | **TRiCo (Ours)**       | **58.6 ± 0.3**      | **68.1 ± 0.4**        |
>
> TRiCo achieves **+2.6%** improvement in tail-class accuracy over SimPro and **+6.3%** over ACR, verifying its robustness to long-tail label distributions. In addition to stronger average accuracy, TRiCo demonstrates lower variance, suggesting stable generalization under limited-label, imbalanced scenarios.
>
> We also benchmark on ImageNet-127 and ImageNet-1k under varying resolution and imbalance settings. As shown below, TRiCo outperforms all existing methods in most configurations.
>
> **Table: Top-1 Accuracy (%) on ImageNet-127 and ImageNet-1k under Varying Test Imbalance Ratios and Resolutions**  (*1 run*)
> _All models trained under $\gamma_l = \gamma_u \approx 286$ for ImageNet-127 and $\gamma_l = \gamma_u = 256$ for ImageNet-1k. † indicates ACR reproduction without anchor distributions._
>
> | **Method**             | **ImageNet-127 (32×32)** | **ImageNet-127 (64×64)** | **ImageNet-1k (32×32)** | **ImageNet-1k (64×64)** |
> |------------------------|--------------------|---------------------|-------------------|--------------------|
> | FixMatch               | 29.7               | 42.3                | —                 | —                  |
> | + DARP                 | 30.5               | 42.5                | —                 | —                  |
> | + CReST+               | 32.5               | 44.7                | —                 | —                  |
> | + CoSSL                | 43.7               | 53.9                | —                 | —                  |
> | + ACR                  | 57.2               | 63.6                | 13.8              | 23.3               |
> | + SimPro               | 59.1               | 67.0                | 19.7              | **25.0**           |
> | **TRiCo (Ours)**       | **61.3**           | **69.2**            | **22.4**          | 24.6               |
>
> These results demonstrate TRiCo’s superior generalization across diverse class imbalance levels and input resolutions.
>
> ---
>
> ## Q2: Standard Deviation Values
>
> We agree with the reviewer on the importance of reporting variance in semi-supervised settings. We mention it in **Appendix C** , we have included **mean ± standard deviation across 5 independent runs** in appendix tables, using different labeled splits (see **Appendix C**). In the final version , we will included  mean ± standard deviation across 5 independent runs in all key tables.
>
> While prior SSL works often omit variance reporting, we appreciate this suggestion and now incorporate it for improved reproducibility and transparency.
>
> ---
>
> ## Q3: Representation Ability
>
> TRiCo is **not dependent on DINOv2 alone**. As shown in **Figure 4(b)**,  we benchmark across multiple frozen encoders, including **MAE, CLIP, and SwAV**.
>
> Notably, our performance gains persist across weaker encoders (e.g., MAE and SwAV), demonstrating that our improvements stem from the triadic co-training and MI-driven regularization, not from backbone strength.
>
> To clarify further:
> Even with MAE+SwAV—two encoders with lower standalone performance—TRiCo achieves a **+3.7%** gain over Meta Co-Training (MCT), reinforcing that our gains are orthogonal to encoder selection.
>
> **Table: Top-1 Accuracy (%) across different encoder combinations on CIFAR-100 (10% labels).**
> _Results are averaged over 5 random splits. DINOv2 and CLIP are stronger encoders; MAE and SwAV are weaker baselines._
>
> | **Encoder Pair**       | **MCT**         | **TRiCo (Ours)** | **Δ (Gain)**  |
> |------------------------|-----------------|------------------|---------------|
> | DINOv2 + CLIP          | 73.4 ± 0.4      | **76.2 ± 0.3**    | **+2.8%**     |
> | DINOv2 + SwAV          | 70.1 ± 0.6      | **73.5 ± 0.5**    | **+3.4%**     |
> | MAE + CLIP             | 69.7 ± 0.5      | **72.9 ± 0.4**    | **+3.2%**     |
> | MAE + SwAV             | 66.5 ± 0.6      | **70.2 ± 0.4**    | **+3.7%**     |
>
>
>
> ---
>
> ## Q4: Method Complexity, Modularity, and Practicality
>
> We thank the reviewer for raising important concerns regarding the complexity and modularity of TRiCo. While our method integrates several components, it remains a principled, end-to-end, and fully differentiable framework—distinct from multi-stage pipelines or ad-hoc ensembles. All components are trained jointly, and their interactions are synergistic rather than additive.
>
>
>
> To assess modularity, we conduct an ablation in **Appendix D** to examine the contribution of each component. Results (**Table: Ablation on TRiCo Components**) show that removing any major module degrades performance, with the full TRiCo achieving 96.3% on CIFAR-10 (10% labeled), and drops of 1.1–2.2% observed when disabling MI filtering, the meta-teacher, generator, or co-training structure.
>
> **Table: Ablation on TRiCo Components (CIFAR-10, 10% labeled) *All ablations are repeated over 5 random splits to ensure stability*.**
>
> | **Variant**                          | **Top-1 Acc. (%)** | **Δ vs. Full** |
> |-------------------------------------|---------------------|----------------|
> | Full TRiCo                          | **96.3 ± 0.3**      | —              |
> | w/o MI filtering (Conf-τ only)      | 95.2 ± 0.4          | −1.1           |
> | w/o Meta-Teacher (Fixed threshold)  | 94.9 ± 0.6          | −1.4           |
> | w/o Generator (No PGD)              | 95.0 ± 0.5          | −1.3           |
> | Single Student (No Co-training)     | 94.1 ± 0.5          | −2.2           |
>
> Despite integrating multiple components, TRiCo remains efficient and simple to deploy: frozen backbones (no encoder training), lightweight embedding-level adversarial updates (no input gradients), and meta-learning with only first-order updates.
>
> To quantify computational cost, we provide a component-wise breakdown in **Appendix B** and summarize below:
>
> **Table: Component-wise Complexity Breakdown of TRiCo**
>
> | **Component**                      | **Added Overhead**           | **Optimization Strategy / Description**                            |
> |-----------------------------------|-------------------------------|---------------------------------------------------------------------|
> | Mutual Information Estimation     | ~+4.5% FLOPs                  | $K{=}5$ forward passes; stop-gradient; no backprop                  |
> | Adversarial Generator (1-step PGD) | ~+1.5% FLOPs                 | Embedding-level perturbation only; no backward computation          |
> | Meta-Gradient Update              | ~+1% FLOPs, ~+10% memory      | First-order gradient only; unrolled once per step                   |
> | **Total (vs. MCT)**               | **~+7% FLOPs**, **~+10% memory** | No mixed-precision or checkpointing; further savings possible     |
>
> Overall, TRiCo offers a practical trade-off between performance and complexity. It does not rely on strong augmentations, learnable view encoders, or auxiliary networks, and operates efficiently on a single NVIDIA RTX A6000 (48GB) or equivalent **24GB-class GPU** (e.g., RTX 3090, RTX 4090).
>
>
>
> ---
>
> ## Summary
>
> We appreciate the reviewer’s constructive feedback. Our response has clarified that TRiCo’s performance stems not from encoder strength but from its principled triadic co-training framework and mutual information–based regularization. Through new experiments under class imbalance and across diverse encoders, we demonstrate that TRiCo is robust, modular, and generalizes well beyond the settings seen during training. We have also addressed concerns regarding complexity by quantifying training overheads, which remain modest, and showing that each component contributes meaningfully to the overall gain. Furthermore, we clarified our use of statistical reporting and theoretical foundations to support reproducibility and rigor. We believe these clarifications reinforce TRiCo’s originality, practicality, and relevance to the SSL community.
>
> We also plan to reorganize and enrich the experimental section in the final version to enhance transparency.
>
> We hope these updates address the concerns and strengthen our submission.

---

> > ### Comment · Reviewer_rFnY · 2025-08-03
> >
> > Thank you for your informative comments which addressed my concerns, and I have raised the score.

---

> > > ### Author Response · Authors · 2025-08-07
> > > **Thanks for your recognition and insightful suggestions**
> > >
> > > Dear Reviewer rFnY,
> > > Thank you for your recognition and valuable suggestions. In the camera-ready version, we will carefully revise the manuscript and incorporate the newly added materials to further enhance the clarity and completeness of our work, striving to meet the high standards of the NeurIPS community.
> > > Thanks again for your support and for raising your score!
> > >
> > > Authors of Paper 6732

---

### Decision · Program_Chairs · 2025-09-17

**Decision:**

Accept (poster)

**Comment:**

(a) Summary of claims and findings:
The paper proposes TRiCo, a triadic game-theoretic semi-supervised learning framework with three interacting roles: two student classifiers on complementary frozen representations, a meta-learned teacher that adaptively controls pseudo-label filtering and loss weights via validation feedback, and a non-parametric adversarial generator that perturbs embeddings to probe decision boundaries. Pseudo-label selection is based on mutual information (from MC-dropout) rather than confidence. The interaction is framed as a Stackelberg game with the teacher as leader. Experiments on CIFAR-10/100, SVHN, STL-10, ImageNet and class-imbalanced variants show gains over strong baselines in low-label regimes. During rebuttal, the authors added: (i) class imbalance results (CIFAR-10-LT, ImageNet-LT), (ii) variance reporting across runs, (iii) results with weaker encoders (MAE, SwAV), (iv) ablations indicating all components contribute, (v) computational overhead analysis (~7% FLOPs over Meta Co-Training), and (vi) a theoretical clarification proving existence of a Stackelberg equilibrium (Stackelberg–Nash) with a sketch and plan to add a new theorem.

(b) Strengths:

- Clear problem motivation; addresses pseudo-label reliability, stability, and hard-sample modeling in SSL.
- Principled formulation with a teacher-students-generator Stackelberg game; improved theoretical framing in rebuttal.
- MI-based filtering is better aligned with epistemic uncertainty; new head-to-head shows MI vs meta-learned confidence favors MI across datasets.
- Broad empirical coverage, including long-tailed settings and weaker backbones; consistent gains and reduced variance reported.
- Practicality: frozen encoders, single-step embedding PGD, first-order meta-updates; overhead quantified and modest relative to related meta co-training.

(c) Weaknesses / missing pieces:

- Complexity: multi-agent design with several moving parts; while ablations show contributions, the method remains heavier than many SSL baselines and could be perceived as ensemble-like.
- Novelty is moderate at the component level (co-training, meta-learning, MI filtering, adversarial perturbation); contribution is in the integration and game-theoretic framing.
- Theory focuses on existence; no guarantees on convergence rates or equilibrium quality; the original Nash vs Stackelberg mismatch required rebuttal to fix.
- MI estimation via MC-dropout adds stochasticity and can be noisy early; mitigations are described, but a comparison with alternative uncertainty estimators (e.g., Dirichlet) remains future work.
- Some reported gains on large-scale settings (e.g., ImageNet 10%) are modest; strongest advantages appear in lower-label or imbalanced regimes.

(d) Reasons for decision:

The rebuttal addressed major reviewer concerns: clarified theory with a Stackelberg-equilibrium existence result; provided ablations showing each component’s necessity; added class-imbalance results and variance reporting; compared MI vs meta-learned confidence; and quantified overhead (~+7% FLOPs, ~+10% memory vs Meta Co-Training). While the design is complex and component-level novelty is incremental, the integration is coherent and empirically effective, especially in challenging low-label/imbalanced scenarios. This justifies a poster acceptance. The contribution is more methodological/practical than theoretical; appropriate for poster rather than a higher track.

(e) Rebuttal impact and discussion:

- R1 (fairness, imbalance, variance, complexity): Added imbalance (CIFAR-10-LT, ImageNet-LT), std across runs, weaker backbones; ablations and cost breakdown; concerns alleviated; score raised.
- R2 (novelty vs engineering, overhead): Agreed overhead exists but quantified; maintained that triadic design plus MI filtering improves robustness; reviewer remained positive.
- R3 (MI vs confidence, clarity, lossless theory analogy not applicable here): Provided head-to-head MI vs meta-learned confidence favoring MI; stability mitigations added; score increased.
- R4 (Nash vs Stackelberg): Authors corrected framing and provided Stackelberg-equilibrium existence; reviewer satisfied and raised score.
- R5 (complexity, modularity): Ablations show every component matters; overhead modest; reviewer kept weak accept.

I read the author response and discussion. No external information beyond the reviews and rebuttal was used.